# Quantifying the economic effects of different fishery management regimes in two otherwise similar fisheries

**Christopher Liese, Scott Crosson** [ORCID] *

NOAA Southeast Fisheries Science Center, Miami, Florida, United States of America

* scott.crosson@noaa.gov

**Data Availability Statement:** NOAA landings data are considered confidential, including the economic information. However, properly aggregated data can be found at https://repository.library.noaa.gov/.

## Abstract

In the southeast U.S., two very similar fisheries are managed by very different management regimes. In the Gulf of Mexico Reef Fish fishery, all major species are managed by individual transferable quotas (ITQs). The neighboring S. Atlantic Snapper-Grouper fishery continues to be managed by traditional regulations such as vessel trip-limits and closed seasons. Using detailed landings and revenue data from logbooks together with trip-level and annual, vessel-level economic survey data, we develop financial statements for each fishery to estimate cost structures, profits, and resource rent. By comparing the two fisheries from an economic perspective, we illustrate the detrimental effects of the regulatory measures on the S. Atlantic Snapper-Grouper fishery and quantify the difference in economic outcomes, including estimating the difference in resource rent. We find that the choice of fishery management regime shows up as a regime shift in the productivity and profitability of the fisheries. The ITQ fishery generates substantially more resource rents than the traditionally managed fishery; the difference is a large fraction of revenue (~30%). In the S. Atlantic Snapper-Grouper fishery, the potential value of the resource has almost completely dissipated via lower ex-vessel prices and hundreds of thousands of gallons of wasted fuel. Excess use of labor is a lesser issue.

## Introduction

Fisheries management takes a variety of different forms, but the most important distinction from a resource economics perspective is the choice between controls to manage the fishery. In traditional U.S. fisheries management, total commercial harvest is indirectly managed by limiting how many vessels are able to harvest the resource by limiting the number of licenses, implementing maximum per-trip catch levels, opening and closing seasons for different species, requiring or forbidding particular types of gear, and other controls of fishing effort [1]. These approaches to management effectively raise the cost of effort in order to limit it. In contrast, fisheries management with catch shares directly manages harvest by dividing the total harvest into separate quotas that exclusively harvested by individuals or groups with much fewer restrictions on fishing effort. As a result, fishers can focus on economic efficiency, i.e., maximizing revenues while minimizing costs. Hence, while both forms of regulation can

**Funding:** The author(s) received no specific funding for this work.

**Competing interests:** The authors have declared that no competing interests exist.

achieve biological goals (resource sustainability), catch shares are much better at capturing economic value for society from a renewable resource [2–4].

Among the most economically efficient of catch shares are individual transferable quotas (ITQs) [5, 6]. Economists have demonstrated conceptually and empirically that introducing ITQs into fisheries rationalizes them, i.e., reduces excessive fishing effort and redundant investment thereby increasing productivity and hence economic profitability. In the U.S., fisheries that have been able to increase profitability following the introduction of ITQs include Alaskan halibut [7], West Coast groundfish [8], and Gulf of Mexico red snapper [9].

In the southeast U.S., two otherwise very similar fisheries operate under two very different management regimes. Their similarity in all but management allow us to explore the divergent economic behavior and outputs that derive from them. The South Atlantic Snapper-Grouper fishery and the Gulf of Mexico Reef Fish fishery are geographically adjacent, utilize the same vessel and gear, catch the same species complex, are integrated into the same regional and national markets for inputs and outputs, and are both under U.S. federal management oversight (and the same data collection regime). However, the two different regional fisheries management councils have taken completely different approaches to management. The Snapper-Grouper fishery is intensely managed by traditional regulations while the Reef Fish fishery has been mostly transitioned to ITQs.

We wish to add to the empirical literature by carefully comparing these two fisheries from an economic perspective. Using detailed landings and revenue data from logbooks together with trip-level and annual, vessel-level economic survey data, we develop financial statement for each fishery to estimate cost structures, profits, and resource rent. These detailed economic measures allow us to contrast the economic performance of these fisheries and to illustrate the detrimental effects of the regulations on the S. Atlantic Snapper-Grouper fishery; as well as to quantify the difference in economic outcomes, including estimating the difference in resource rent generated. We take a slightly different perspective from much of the empirical literature on resource rent, in that our primary focus is on the inefficiencies brought about in the traditionally managed fishery rather than focusing on the efficiency gains of a fishery that transitioned to an ITQ [10].

## Materials and methods

### Discussion of framing of rents / brief literature review

Here, we define a commercial fishery that has achieved a state where overall fishing activity only covers business costs and opportunity costs as lacking resource rents [11]. We define any profits beyond business and opportunity costs as fishery rents. Our analysis below will use these definitions to illustrate the impact of traditional regulations and ITQs on two similar fisheries.

The ability of an economy to extract resource rents from fisheries has long been a concern of resource economists. Businesses tend to focus on profits, but some profits are expected even in industries that lack significant barriers to entry (beyond the capital and skills required to enter a market). Many economists that study fisheries assume that for society to maximize the value of natural resources—a "gift of nature"—those resources should be extracted in a way that minimizes costs [12]. If a natural resource is privately owned, that may well occur, but commonly owned resources are a different story [13].

There are several significant barriers to extracting resource rents from commonly owned renewable resources (as most all wild caught fisheries are). The first is the long established susceptibility of fishery stocks to overexploitation [14]. This is a standard commons-type problem, with individual fishermen perhaps not intending to extract from the stock to the point of

diminishing returns, but the lack of incentives to manage jointly leading to that result overall. To a large extent, the point of fishery management is to prevent that overexploitation from occurring [15].

Unfortunately, a second barrier to extracting potential rents from fishery stocks may well be the fisheries management process itself. There are a number of potential ways to limit over-fishing by commercial fishermen. Limiting entry is certainly one of those, but even a limited-entry fishery will not necessarily lead to an efficient outcome because the commons problem still exists for the current participants [1].

In fact, regulated open access can lead to additional inefficiencies [16, 17]. Additional management measures may include putting up obstacles to rational fishing behavior, such as limiting the harvest of particular species on each vessel or trip, limiting the use of the most efficient fishing gears, limiting fishing seasons, creating closed areas, and other restrictions that managers develop to make the business of fishing less profitable and hence less palatable. An unregulated fishery will eventually drive down harvest to the point that fishermen will only pursue the species to the point of covering their business costs and opportunity costs. A regulated, limited entry fishery may do the same, although at a higher harvest level and with better protection of the underlying stock [18, 19]. In either case, the regulations restrict options and hence lead to inefficiencies.

Contrasting systems have many names: rights based, market-based, rationalized, catch shares, ITQs, individual fishing quotas. They have in common that harvest is exclusively assigned to certain actors, thereby deterministically achieving a set quota (short of cheating, of course). The resource economics literature on the effects of ITQs on fisheries is vast. Researchers have noted the effects of ITQs on sustainability [20], communities [21], safety [22], equality [23], and diversification [24]. Some of the social effects of ITQs can be negative, but our intention here is not to add another case study of the effects of ITQs, but instead to demonstrate the inefficiencies of standard regulations by contrasting them with a neighboring alternative rights-based management system.

## The fisheries of the snapper-grouper species complexes in the Southeast U. S.

The U.S. federal law decentralizes much of its fisheries management by handing management to regional fishery management councils (FMC). While the National Oceanic and Atmospheric Administration (NOAA) and its parent the U.S. Department of Commerce retain the ultimate responsibility for preventing overfishing, the FMCs debate and choose most of the actual regulations after consulting with their scientific advisory committees [25]. In the southeast U.S., the South Atlantic Fishery Management Council (SAFMC) and Gulf of Mexico Fishery Management Council (GFMC) are responsible for federal fisheries management. The SAFMC is responsible for managing fishing in federal waters from the North Carolina-Virginia border to the bottom of the Florida Keys, and the GFMC oversees the federal waters from the Keys border to the Texas border with Mexico (Fig 1). The two areas are mostly at the same latitudes (although the SAFMC jurisdiction stretches further north), share similar weather (and hurricanes), are culturally similar, and are split by the Florida peninsula. The state of Florida is represented on both the SAFMC and GFMC. Most commercially valuable species found in one area are also present in the other, although the size of the stocks varies due to topography and ecology. The Gulf of Mexico area is both larger and contains a larger, shallower continental shelf.

In both regions, commercial fishermen harvest the snapper-grouper species complex often associated with bottom structures; also known as reef fish. Species include over 40 different

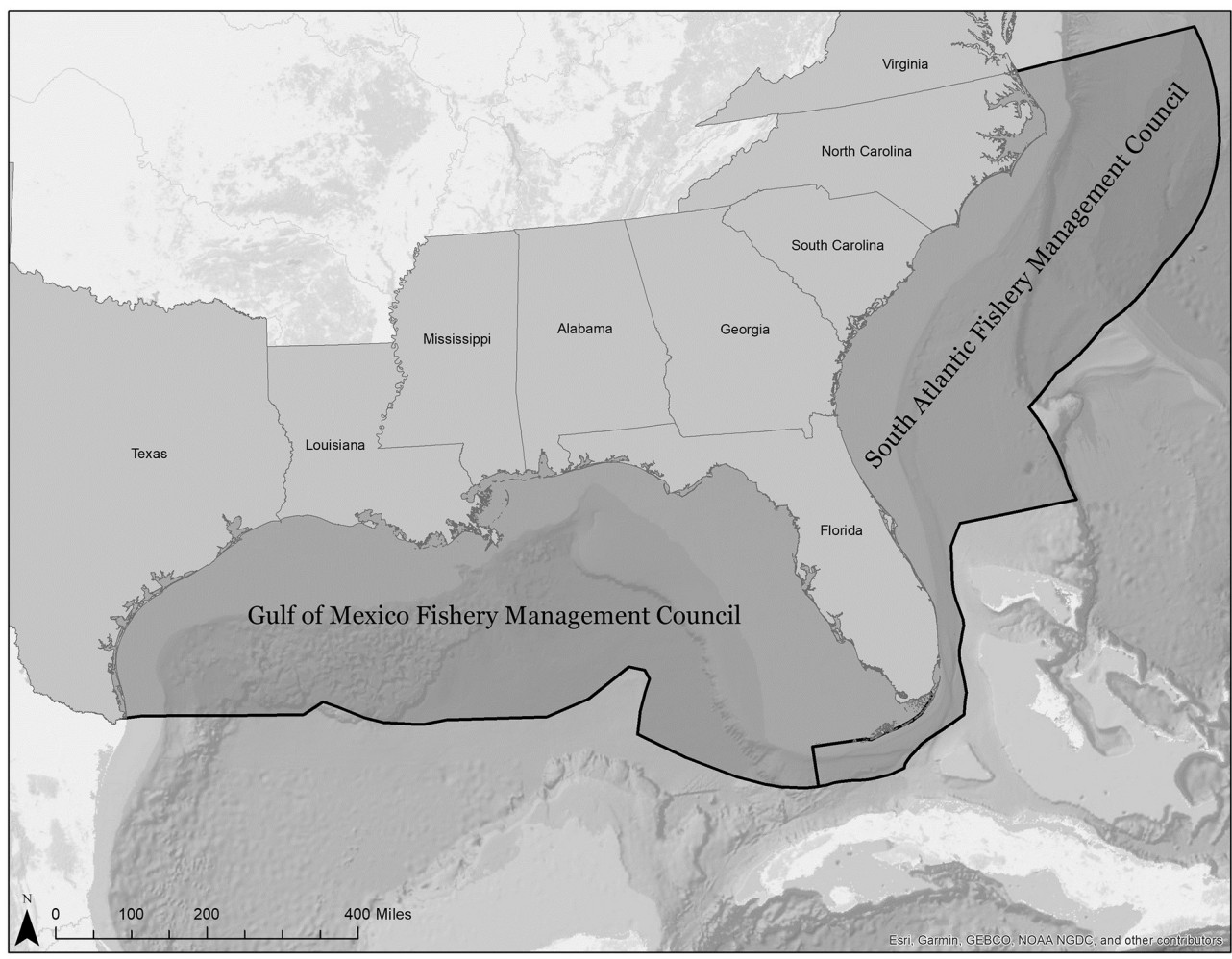

**Fig 1. Waters of the GFMC and SAFMC.** Map created by Megan Slemons, Emory Center for Digital Scholarship.

snappers, groupers, tilefishes, and others. These fish are not the economically dominant species in either region as shrimp, lobsters and crabs, as well as other finfish, e.g., menhaden, make up the large majority of the seafood produced in lbs and dollar terms [26].

The commercial reef fish fisheries in both regions are also conducted in a similar manner. Vessels are very similar—around 35 feet in total length, built of fiberglass in the late 1980s, with about 400 horsepower engines, and using ice as refrigerant. They use the same type of gear, predominantly vertical lines, including hand lines, electric reels, and bandit gear. The next biggest gear groups in both regions are bottom longlines and diving. Also, very important from an economic perspective, the two fisheries are embedded in the same markets. The markets for fish extends throughout the region and the nation, with dealers regularly shipping to NY. Also, fuel, labor, gear, and vessels are all sourced from the same southeast U.S. market, with prices matching and fluctuating together.

In fact, the two fisheries are so similar, the managers and fishery scientists refer to these two fisheries by different names to help keep them apart. In the SAT, the species complex is usually referred to as snapper-grouper while in the GOM this complex is called reef fish. For the remainder of the paper, we will follow this convention and abbreviate "South Atlantic Snapper-Grouper" as SAT SG and "Gulf of Mexico Reef Fish" as GOM RF.

Managing these reef fish fisheries is biologically and technically complex. It is biologically complex because many aspects of their life cycles make them particularly susceptible to over-fishing. Some species live for decades and do not mature until they are several years old, and some are sequential hermaphrodites (beginning life as males and maturing into females later in life). Fisheries that disproportionately target larger fish, either because of desirability or regulations (i.e., minimum size limits), will disproportionately affect the reproductive capacity of the entire stock. Another complication is that reef fish that are caught in waters deeper than approximately 100 feet will suffer barotrauma when quickly brought to the survey. This increases the discard mortality for caught fish that are released due to being the wrong size or out-of-season species (regulatory discards). This problem is compounded by the inability of fishermen to selectively target many of the individual species as they cohabitate the same eco-system. Most reef fish are also considered tasty fare and in high demand from both commercial and recreational fishermen. Overall, reef fish stocks are easy to target and easy to overfish, difficult to select for, face a high discard mortality rate, and are easier to find in recent decades due to the advent of fish finders and GPS technology (storing and finding reefs or structure precisely).

The GMFC and SAFMC have faced these challenges simultaneously. Beginning in the late 1980s, federal fisheries management in the U.S. began moving from an emphasis on commercial harvesting to biological sustainability [25]. While the GFMC Reef Fish Fishery Management Plan and the SAFMC Snapper-Grouper Fishery Management Plan both date to the mid 1980s, the Councils only began seriously regulating, i.e., restricting, the reef fish fisheries in the 1990s. This included the passage of various size limitations and then a new commercial license moratorium passed by the GFMC in 1996 and the SAFMC in 1997, limiting access to new fishers. The SAFMC started the first ITQ for finfish in the U.S. in 1993 for the wreckfish fishery, but has not passed another since [27].

After experiencing many of the problems with ever increasing regulations (short mini-seasons, derby fishing, high size limits and much discarding), the GFMC started its first ITQ for red snapper in 2007 [9], and has since started another for 13 species of groupers and tilefish [28, 29]. With the introduction of ITQ for red snapper, the GFMC removed the management by seasons and closures but also reduced the size limit, and instituted a big quota cut (which was reversed fairly quickly as the stock recovered). The majority of revenue from the GOM RF fishery now comes from species managed under ITQs, as we will describe below. In contrast, the SAT SG fishery is managed entirely with traditional regulations.

Golden tilefish provides a useful practical example of the difference between the two management regimes in the two adjoining regions. In the GOM RF fishery, the commercial quotas for ITQ-managed species are annually converted into pounds and split among the quota owners proportionately to their share of the total quota, so a quota owner with 1% of the quota for tilefish, for example, is eligible to catch poundage equivalent to that portion of the overall quota or (alternatively) lease it to another vessel for its use. All commercial vessels harvesting tilefish thus need a limited-access reef fish permit and can either lease tilefish quota allocation (valid for a year) or permanently purchase tilefish shares (that annually spawn quota allocation) from others in the ITQ program (if they were not allocated them at the start of the program). The quota allocation applies to any of the tilefish species caught on the trip, and non-tilefish species co-caught on the trip (generally yellowedge grouper) will require a separate quota of its own (deepwater complex quota). There are some gear restrictions, but most vessels will use the most efficient gear (bottom longline) as the species lives in deeper waters. The fishery is year-round, and vessels can time their trips to balance the freshness of the harvest with expected demand and the variable and opportunity costs incurred on trips of varying lengths.

On the other side of Florida in the SAT SG fishery, the commercial tilefish quota is divided into a hook and line component (for all fishers possessing a limited entry snapper-grouper permit) and a longline component (for fishers possessing one of 23 vessels possessing an additional golden tilefish endorsement). Regulations are designed to slow the overall harvest. The hook and line catch of golden tilefish is limited to 500 lbs per trip, and when that portion of the quota is met, the fishery is closed until the following year. In 2020, that portion of the fishery closed on July 23rd. This is still longer than the longline component, as their fishery was closed on February 18th as a precautionary measure; despite a 4000 lbs trip limit. After the quota was found not to have been met, their season was reopened for nine additional days on March 14th, then closed until January 2021. Golden tilefish from the Gulf is available year round, but the same species is only available until mid-year from the South Atlantic, and suffers from pulses and surges as fishermen race to catch as much as possible before the closures begin.

## Ethics statement

Economic surveys of commercial fishermen were approved by the United States Office of Management and Budget (OMB) as part of NOAA's regulatory authority. As with landings data, survey results tied to individual fishermen are considered confidential data and can be released only after aggregating to include at least three participants. Aggregated data can be found at the NOAA repositories at https://repository.library.noaa.gov/.

## Data

For our comparative analysis, we use fisheries logbook and economic data from NOAA's National Marine Fisheries Service's (NMFS) Southeast Fisheries Science Center (SEFSC). Since 1993, the SEFSC has required all fishing vessels to report on their commercial fishing activity for federally managed species, including the SAT SG and GOM RF fisheries. Fishers must complete and submit a trip report (logbook) for every fishing trip to remain compliant with their federal fishing permits. The fishing logbooks are nearly a complete census of landings and effort in the federally managed commercial fisheries in the southeast U.S. To estimate revenue at the trip level (i.e., for each logbook), we multiply the logbook's landings poundage by the most appropriate price available from the dealer landings data summarized in the SEFSC's Accumulated Landings System (ALS). An algorithm matches dealer, state, month, and year between logbooks and ALS records for each species at the highest resolution possible.

Since 2006, SEFSC economists have conducted two economic surveys to collect economic data at both the trip-level and annual, vessel-level to complement the logbook data. The economic surveys are designed to provide data that in turn can produce fishery-level financial statements; to measure and track the economic developments in these federally-managed fisheries.

Each year, a random stratified sample of permitted vessels is selected to provide trip-level economic information. Selection eligibility is based on whether a vessel has a valid federal permit of interest during late November of the previous year. Approximately 30% of active vessels and 10% of inactive vessels are randomly sampled. For each fishing trip, selected vessels must complete the trip expense section located at the bottom of the trip report form. These variable cost questions include expenses for bait, ice, groceries, and IFQ leasing (paying to use another vessel's shares); the amount of fuel used and the cost per gallon of fuel; whether or not the vessel owner was present on the trip, and whether or not payment for the catch was determined. If payment was determined, then gross trip revenue and payment to hired crew and hired captain are collected (Fig 2).

**Fig 2. Trip-level economic survey instrument (part of logbook report).**

Early in the following year, selected vessels are mailed an annual expense survey (S1 Fig). This survey elicits annual, vessel-holistic economic data. The primary purpose of the annual expense survey is to collect fixed costs. These expenses include the costs for maintaining and repairing the vessel and gear, insurance, loan payments, and overhead (such as mooring, utilities, office staff, professional services, etc.). Because vessels often engage in (non-federal) fisheries not in the logbook system, or engage in for-hire fishing, the survey also asks for annual cumulative trip-level expenses such as fuel, supplies, and hired crew payments for a complete picture of the annual, vessel-level expenses. To allow for comparison to the trip-level reporting and to help assign shares of fixed costs, the survey also collects the number of days at sea and total revenue for commercial and for-hire fishing. Finally, it collects an estimate of the vessel's market value as proxy for the capital invested.

The surveys involve three rounds of mail-outs, reminder calls, many call-backs, and send-backs when call-backs fail. Response rates are generally high by fishery standards. In 2016, the raw response rate at the trip-level was 98% and 99% in the SAT SG and GOM RF fisheries, respectively. The annual, vessel-level raw response rate was 77% and 88%, respectively.

Many call-backs to respondents are necessary for getting missing values, clarifications, or validating outlier or odd numbers. Only records that are complete in all the financial fields can be used for generating the financial statements, i.e., item non-response cannot be tolerated for most questions. As a result, when counting only observations used in the analysis, the effective response rates drop to 94% and 94% at the trip-level and 71% and 82% at the annual, vessel-level for the SAT SG and GOM RF fisheries, respectively. Missing trip-level revenue or hired crew costs (these questions cannot always be answered by all fishers at the time the logbook is completed) are replaced with the estimated revenue from the logbook and a regression-based estimate of crew costs, respectively. Once the economic data is complete, a careful accounting exercise begins.

Two economically relevant values are not collected on the surveys and are instead estimated for each trip or vessel. At the trip-level, the opportunity cost of the owner-operator's time as captain is estimated based on hired crew compensation and profitability (due to share systems). At the vessel-level, vessel depreciation is simply calculated at 5% of the vessel's current market value. The resulting number is a rough estimate, identical to that used in the federal Gulf of Mexico shrimp fishery where depreciation is based on surveys and the fact that the Internal Revenue Service requires non-fishing vessels to be depreciated over 23 years.

This paper's comparative analysis of the economics of the SAT SG and GOM RF fisheries builds on earlier technical memoranda [30, 31]. Each report provides economic results for a specific subsets of the overall logbook and survey data, such as for the SAT SG or GOM RF fishery. Participation in a fishery is defined as catching at least one pound of the applicable species on a trip in the applicable waters, e.g., in the SAT or GOM. As sampling is at the vessel-level, prior to the fishing year, post-stratification is used to statistically estimate appropriate population means for the elements in the fishery financial statements (separately at the trip-

level and at the annual, vessel-level, as they are effectively two different data streams). All vessel and logbook trip data utilized in this report were pulled from the various databases on May 4, 2018. All dollar values are in nominal 2016 USD.

## Results

Aggregating the census-level logbook data specific to each fishery reveals some important distinctions between the patterns of the fishing in the two regions (Table 1). The first is the relative size of the each fishery as the total landings in gutted-weight pounds in the GOM RF fishery are nearly three times those in the SAT SG one. The Gulf of Mexico is a larger area in terms of geography and habitat as the reef species are caught on the continental shelf and preferably in shallower waters. To account for the difference in scale of these two fisheries, we calculate the ratio of measures in the SAT SG fishery over the GOM RF fishery and then adjust for the difference in pounds (divide by 0.35), i.e., we prorate appropriately for a more legitimate comparison.

For instance, revenues are higher in the GOM RF fishery and not just due to higher landings. The data shows that on average fishers in the GOM RF fishery received a higher price per pound (+$0.74). This implies that SAT SG fishers receive just 82% of revenue per pound of fish landed compared to the GOM RF fishers. Despite its much smaller catch, the SAT SG fleet is approximately the same size as the GOM RF one. When adjusted for pounds caught, the SAT SG fleet contains almost three times as many vessels as the GOM RF fleet. Even starker is the divergence in the number of trips taken in each fishery. For a given amount of landings, the SAT SG fleet takes almost five times more trips than the GOM RF one. It is also worth noting that the SAT SG fishery uses 29% more labor per unit of landings than the GOM RF fishery.

Looking at 2016 fishing trips (Table 2), we see that GOM RF trips are much longer; averaging 4.4 days at sea compared to 1.7 days of SAT SG trips. On these longer trips, with a somewhat larger crew, Gulf fishers on average land more than four and a half times as much, 2,262 pounds vs. 499 pounds, as fishers in the SAT SG fishery. Trip limits in the SAT SG fishery are leading to much shorter trips than would otherwise be taken. Regulations for most of the major commercial fishing species in the SAT SG fishery are intended to extend fishing opportunities, i.e., preventing very short seasons, by imposing trip limits.

Fig 3 shows two scatter plots of trips landing vermilion snapper, a major species, especially in the SAT SG fishery. The top and bottom plots are for the GOM RF and SAT SG fisheries,

**Table 1. Census-level aggregate data for the South Atlantic snapper-grouper (SAT SG) and Gulf of Mexico reef fish (GOM RF) fisheries (annual averages for the period 2014–2016).**

|  | SAT SG | GOM RF | SG/RF Ratio | SG/RF Ratio (per lb basis) |
|---|---|---|---|---|
| SG or RF Fisheries |  |  |  |  |
| Landings | 5,341,587 | 15,176,791 | 0.35 | 1.00 |
| Price | 3.29 | 4.03 | 0.82 |  |
| Revenue | 17,559,439 | 61,199,156 | 0.29 | 0.82 |
| Trips | 11,521 | 6,751 | 1.71 | 4.85 |
| Vessels | 518 | 522 | 0.99 | 2.82 |
| Other landings | +11% | +5% |  |  |
| All Landing on SG or RF Trips |  |  |  |  |
| Landings | 5,908,616 | 15,944,854 | 0.37 | 1.05 |
| Revenue | 19,303,965 | 62,494,512 | 0.31 | 0.88 |
| Crew days | 40,565 | 89,035 | 0.46 | 1.29 |

**Table 2. Average fishing trip in 2016 in the South Atlantic snapper-grouper (SAT SG) and Gulf of Mexico reef fish (GOM RF) fisheries (census-level data).**

|  | SAT SG (N = 11,521) | GOM RF (N = 6,751) |
|---|---|---|
| Days at sea | 1.7 | 4.4 |
| Crew size | 2.0 | 2.8 |
| Landings (lbs) | 499 | 2,262 |
| % of Landings in Fishery | 90% | 96% |

respectively. Each trip is plotted relative to the scale of vermilion snapper revenue (X-axis; in $) and the specialization on vermilion snapper of the trip (Y-axis; in % of vermilion snapper revenue of total trip revenue). When comparing the two plots, we note that the one for the GOM RF fishery lacks any major discontinuities, but the one for the SAT SG fishery has two areas where trips group up along a seeming vertical line; at approximately $2000 and $3750 in vermillion snapper revenue. This strong behavioral response caused by the step-down trip limits implemented by the SAFMC to extend the fishing year. Specifically, as the vermilion snapper quota gets closer to being reached during the year, the per-vessel trip limit for vermillion snapper steps down from 1000 lbs per trip to 500 lbs per trip (until the fleet quota is met and the season is closed).

Somewhat less obvious but equally important to note in Fig 3, is that under the GFMC's management system relatively few trips specialize, or solely target, vermillion snapper. For

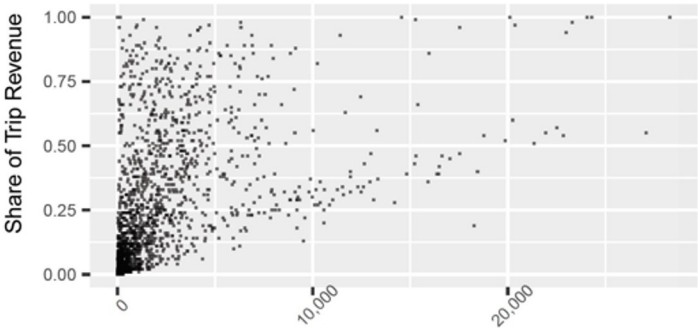

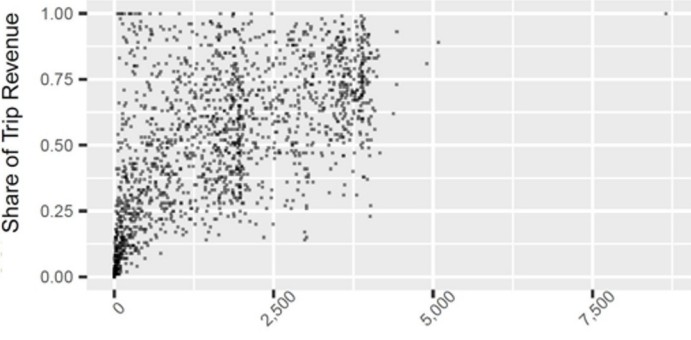

**Fig 3. Scale and specialization of vermilion snapper trips.** Distribution of trips across vermillion snapper revenue (in $) and share of revenue (in %) for the GOM RF (top) and SAT SN fisheries (bottom) in 2016.

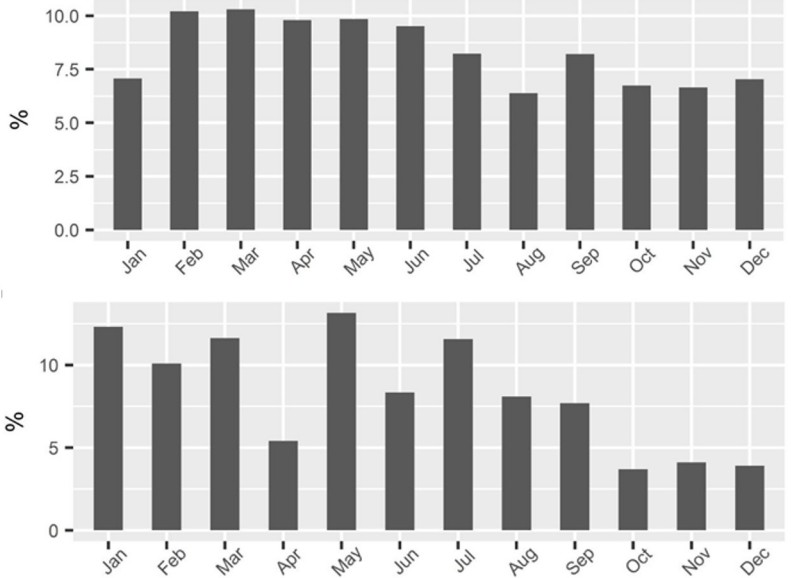

**Fig 4. Monthly share of total revenue in the GOM RF (top panel) and SAT SG (bottom) fisheries in 2016.**

most trips in the GOM RF fishery, vermillion snapper makes up less than a quarter of the revenue for the trip (>70% of trips), with relatively few exceeding half (~15% of trips). In contrast, vermillion snapper trips in the SAT SG fishery derive a much bigger share of their revenue from vermilion snapper. The vermilion snapper revenue exceeds 50% on about 55% of the trips. As will be explained more below, some of the higher specialization in the SAT vs. GOM fishery is likely due to SAT SG season closures of other species.

Fig 4 shows the monthly share of total revenue generated by the two fisheries. The overall harvest level in the GOM RF fishery is much more stable month-to-month than in the SAT SG fishery. In the SAT SG in 2016 months late in the year generate less than a third of the revenue of top producing months, i.e., January and May. In contrast, in the Gulf the lowest month's revenue is still at about two-thirds of the highest month's revenue.

In the SAT SG, a pattern of the fleet running out of quota happens consistently enough across the species that constitute the majority of fishing revenue such that fishers catch noticeably less fish as the winter holiday season approaches. In Fig 4, the last three months of the year, i.e., 25% of the year, constitute only 10% of the year's revenue.

When we disaggregate Fig 4 by species for the SAT SG fishery the impact of seasons, closures, and derby fishing behavior becomes more apparent. Fig 5 shows the monthly share of revenue for three species/species groups. For vermilion snapper (top panel), the quota is split into two seasons, starting in January and July. As each half-year season goes into effect, nearly half of the year's quota is caught in those two first-months after opening. With continued high landings in the following two months quota is quickly drawn down. As a result, the fishery is all but closed in April through June and again October through December. The small spike in landings at the end of 2016 is due to NMFS's oversight of the fishery. After estimating that some quota still remained following the October closure (and data lags), the agency reopened the fishery for a few days before the end of the 2016.

The middle and bottom panel of Fig 5 show the equivalent results for the deepwater species complex and the shallow-water complex of the SAT SG fishery, respectively. The deepwater season opens in January, and two-thirds of landing occur in the first quarter of the year. For

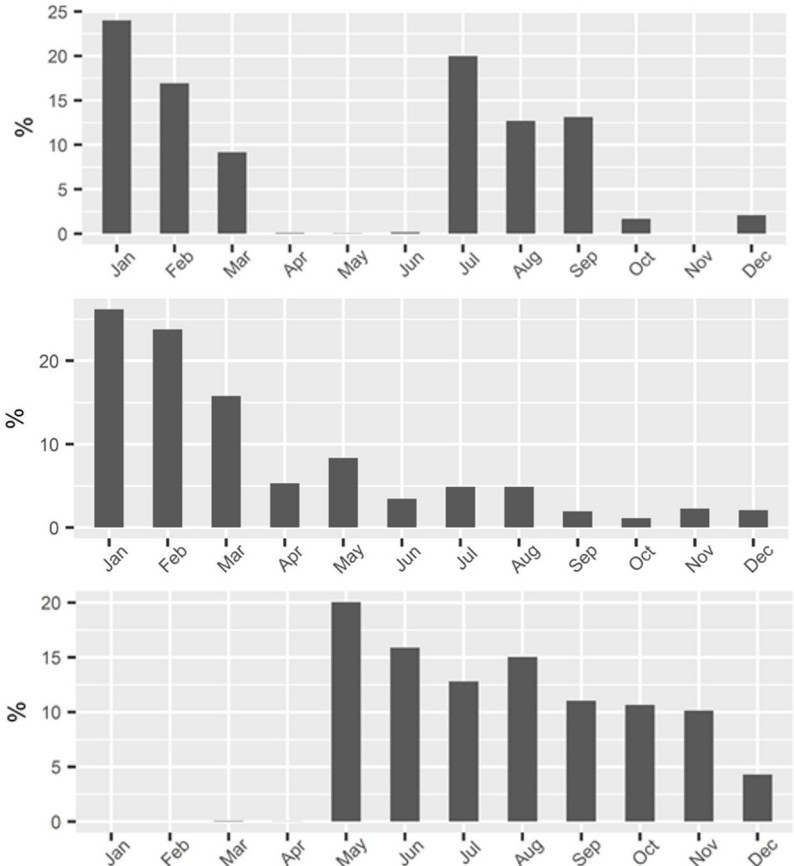

**Fig 5. Monthly share of landings revenue for vermilion snapper (top panel), the deepwater complex (middle), and shallow water complex (bottom) in 2016 in the South Atlantic snapper grouper fleet.**

the shallow-water species, a spawning season closure is in effect for the first four months of every year. The expectation of closures due to exhaustion of the quotas induces some element of derby behavior, i.e., racing to fish behavior, in the SAT SG fishery.

The use of some gear types in the SAT SG fishery correlates fairly strongly with certain species or species groups. For instance, longlines are used to catch half the deepwater complex landings, almost entirely during January through March. Diving equipment and traps are used May through October. As a result, much of this gear is idle for at least half a year. While some idle and redundant gear serves economic purposes, too much is a drain on productivity. It is also likely that the derbies and season closures are detrimental to the average ex-vessel prices fishers receive in the SAT SG fishery. Due to regulation, and unlikely to match market demand, a perishable product floods the market at times, only to disappear entirely at other times of the year (Fig 5). At least some of the 18% lower average price in the SAT SG is likely due to these fluctuations.

Cost data also reveals substantial differences between the SAT SG and GOM RF fisheries. Based on sample trip-level data for 2016, Table 3 shows that the average GOM RF trip was far more lucrative than those taken in the SAT SG fishery; with gross revenues nearly five times as large. After accounting for trip-level economic costs, i.e., variable costs including the opportunity costs of the owner-operators' time as captain, the GOM RF trips generate over eight times the surplus cash flow (here called trip net revenue) over SAT SG trips. While GOM RF trips

**Table 3. Average trip-level economics in 2016 in the South Atlantic snapper-grouper (SAT SG) and Gulf of Mexico reef fish (GOM RF) fisheries (sample data).**

| | SAT SG (n = 2,612) | | GOM RF (n = 1,948) | |
|---|---|---|---|---|
| | **Mean** | **SE** | **Mean** | **SE** |
| Owner-operated | 82% | 3.1 | 68% | 3.5 |
| Days at sea | 1.8 | 0.2 | 4.4 | 0.2 |
| Crew size | 2.0 | 0.1 | 2.7 | 0.1 |
| Fuel used (gallons) | 74 | 7 | 179 | 11 |
| Landings (lbs) | 503 | 57 | 2,043 | 169 |
| **Revenue ($)** | **1,761** | **206** | **8,406** | **757** |
| **Costs ($)** | | | | |
| Fuel | 165 | 15 | 365 | 21 |
| Bait | 126 | 23 | 303 | 31 |
| Ice | 38 | 5 | 143 | 14 |
| Groceries | 62 | 10 | 262 | 21 |
| Miscellaneous | 49 | 16 | 250 | 34 |
| Hired crew | 517 | 84 | 2,277 | 247 |
| Opportunity cost—Owner-captain | 299 | 35 | 630 | 101 |
| **Trip net revenue ($)** | **505** | **74** | **4,176** | **442** |

are nearly three times the length of SAT SG trips and catch four times the landings, these scale measures are insufficient to explain the eight-times disparity in trip net revenues.

We can compare two proxies for the technical productivity of these trips by calculating the average landings pounds per gallon of fuel used and average landings per crew-day of labor employed. The GOM RF trips generate 68% more landings per gallon of fuel use (11.4/6.8) and 20% more landings per crew-day of labor employed (169/141). This indicates that the SAT SG trips are very inefficient in fuel use vs. the GOM RF trips. At the same time, while labor is employed less efficiently than in the Gulf, the difference is much less than the disparity in fuel consumption. This finding is consistent with the idea that SAT SG trip limit regulation force fishers to cut short many trips, burning more fuel as they frequently return to port to unload.

The two fleets have some differences in average trip-level production functions. While the SAT SG trips spend about twice as much (relative to revenue) on fuel and bait compared to GOM RF trips, they spend a very similar portion of revenue on groceries and miscellaneous expenditures. Ice expenditures and overall labor cost (hired crew and owner opportunity costs) as a share of revenue are about 30% higher in the SAT SG fishery. Dividing trip net revenue by total revenue generates the gross margin generated by the trip, i.e., the share of revenue available, after subtracting trip-level variable costs, to pay for fixed costs, for financing costs, for compensating the owner and invested equity, and pure profit such as resource rent. The trip-level margins are 49.7% and 28.7% for the GOM RF and SAT SG trips, respectively. A 21 percentage point difference on a margin or return in similar industries demands further explanation.

The economic trip-level results---while based on a large sample size and being indicative of the economic situation---do not provide a full or holistic view of the economic situation in each fishery. Annual surveys reveal that the average vessel in each fishery also engages in some for-hire fishing work and commercial fishing for species beyond the SAT SG and GOM RF fisheries (Table 4). At the annual level, overall annual fishing days at sea and for-hire days at sea and revenue are similar, as is the vessel value. Costs related to the vessel, repair and maintenance, insurance, depreciation, and even overhead are also of similar magnitude. The biggest

**Table 4. Average annual, vessel-level economics in 2016 in the South Atlantic snapper-grouper (SAT SG) and Gulf of Mexico reef fish (GOM RF) fisheries (sample data).**

| | SAT SG (n = 94) | | GOM RF (n = 121) | |
|---|---|---|---|---|
| | **Mean** | **SE** | **Mean** | **SE** |
| Owner-operated | 89% | 3.4 | 78% | 3.4 |
| Days-Commercial fishing | 80 | 6.3 | 74 | 3.8 |
| Days-For-hire fishing | 10 | 2.9 | 10 | 3.0 |
| Vessel value ($) | 93,685 | 10,395 | 85,688 | 6,327 |
| Has insurance | 45% | 5.3 | 38% | 4 |
| **Total revenue ($)** | **69,373** | **9,014** | **132,167** | **16,043** |
| Commercial fishing | 57,489 | 7,194 | 120,155 | 15,483 |
| For-hire fishing | 11,883 | 5,442 | 12,012 | 3,625 |
| **Costs($)** | | | | |
| Fuel | 7,037 | 717 | 8,907 | 832 |
| Other supplies | 10,015 | 1,277 | 14,263 | 1,152 |
| Hired crew | 19,274 | 2,853 | 32,336 | 3,942 |
| Vessel repair & maintenance | 10,503 | 1,766 | 11,271 | 1,066 |
| Insurance | 1,478 | 265 | 1,347 | 200 |
| Overhead | 7,100 | 974 | 6,800 | 749 |
| Opportunity cost—Owner | 9,052 | 984 | 8,825 | 1,100 |
| Depreciation | 4,684 | 520 | 4,284 | 316 |
| **Net revenue from operations($)** | **230** | **4,328** | **44,133** | **11,310** |

differences are the revenue from commercial fishing (more than double in the Gulf), followed by hired crew and other supply costs.

There are different ways to look at profits. As our objective is a societal, economic perspective (vs. a individual business, financial perspective), we use what we call net revenue from operations. Net revenue from operations starts with operating revenue (i.e., excluding extraordinary, i.e., non-fishing, income) and subtracts all real, tangible costs of production. Beyond material inputs (fuel, repairs, etc.), the in-kind contributions to the production process must be accounted for as well. In our case, this includes the opportunity cost of owner-operator's time spent as captain of the vessel and the vessel's depreciation accounting for the degradation of the vessel with use and over time. Financial cost that do not represent an actual input to the production function, e.g., loan payments, IFQ purchases, or income taxes, are not counted. From a societal perspective many of these represent transfers of value generated by the fishery to others, i.e., are distributional in nature (and not our focus here).

To be more representative of the fisheries than a single year, we collapse the cost categories, express them as percent of revenue, and then average them across three years (Table 5). We thereby generate an aggregate cost structure in percent-of-revenue terms and the economic profit margin implicit in the annual, vessel-level net revenue from operations. We will use these measures for the rest of the paper.

In Table 6, we use the 3-year average cost structure and margins from Table 5, multiplied by the 3-year average annual fishery revenue (from Table 1) to estimate the total annual fishery expenses in each cost category and fishery-wide total profit (net revenue from operations). Note that the SAT SG and GOM RF landings that generate the revenue displayed are never landed in isolation from other species. Similarly, fixed costs components are never specific just to these fisheries. Hence the table's results represent an abstraction (through standardization and prorating) of complex and messy fisheries data down to the a hypothetical concept of a pure SAT SG or GOM RF fishery, respectively.

**Table 5. Three-year average of economic costs and net revenue as percentage of revenue for the South Atlantic snapper-grouper (SAT SG) and Gulf of Mexico reef fish (GOM RF) fisheries (for the period 2014–2016).**

| | SAT Snapper-Grouper | | | | GOM Reef Fish | | | |
|---|---|---|---|---|---|---|---|---|
| | 2014 | 2015 | 2016 | Mean | 2014 | 2015 | 2016 | Mean |
| | (n = 75) | (n = 101) | (n = 94) | | (n = 84) | (n = 105) | (n = 121) | |
| Revenue | 100% | 100% | 100% | 100% | 100% | 100% | 100% | 100% |
| **Costs (% of Revenue)** | | | | | | | | |
| Fuel & Supplies | 27.1% | 24.6% | 24.6% | 25.4% | 18% | 15% | 18% | 17.1% |
| Labor—Hired & Owner | 39.0% | 36.6% | 40.8% | 38.8% | 32% | 31% | 31% | 31.5% |
| Vessel R&M, Insure, Overhead | 23.0% | 25.7% | 27.5% | 25.4% | 14% | 13% | 15% | 14.1% |
| Depreciation | 5.3% | 5.3% | 6.8% | 5.8% | 3.7% | 3.0% | 3.2% | 3.3% |
| **Net Revenue from Operations** | 5.6% | 7.7% | 0.3% | 4.5% | 31% | 38% | 33% | 34.0% |

Conceptually, to derive the resource rent from the net revenue from operations, it is necessary to subtract the opportunity costs of capital. The opportunity cost of capital accounts for "fair" compensation for the financial capital invested in the fishing vessel and business. For private businesses investment decisions, a large element accounts for the investment risk involved. Past studies in fisheries have assumed an opportunity cost of capital which is equal to the rate of return on a BAA rated bond, which is considered a somewhat risky bond [32]. During the 2014–2016 time period, the rate for such bonds averaged 4.85%. For evaluating a publically-owned natural resource at the aggregate industry level, and not to penalize the more capital intensive SAT SG fishery further, we use a more conservative opportunity cost of capital of 3.5%.

We apply this rate to the total market value of the effective vessels in each fishery. To calculate the number of effective vessels, we first use the days at sea (from the annual, vessel-level surveys) to prorate the total number of vessels between commercial and for-hire fisheries. In a second step, we prorate the commercial fishery effective vessels between the fishery of interest (SAT SG or GOM RF) and any other fisheries by using the revenues from the logbooks. In this manner, we calculate that the 518 vessels (partially) active in the SAT SG, are "equivalently engaged" as 272 hypothetical vessels that are fishing—solely—for SAT SG species. We then multiply this number by the 3-year average vessel value ($82,793) to estimate the value of the capital stock ($22.5 million invested capital specific to the SAT SG fishery). 3.5% of this capital stock corresponds to the $0.8 million under opportunity cost of capital in Table 6. The equivalent is done for the Gulf.

**Table 6. Estimated total annual economic costs, net revenue, and resource rent for the South Atlantic snapper-grouper (SAT SG) and Gulf of Mexico reef fish (GOM RF) fisheries (for the period 2014–2016).**

| | SAT SG | | GOM RF | |
|---|---|---|---|---|
| | as % of Rev. | in $ million | as % of Rev. | in $ million |
| **Revenue** | **100.0%** | **17.6** | **100.0%** | **61.2** |
| **Costs** | | | | |
| Fuel & Supplies | 25.4% | 4.5 | 17.1% | 10.5 |
| Labor—Hired & Owner | 38.8% | 6.8 | 31.5% | 19.3 |
| Vessel R&M, Insure, Overhead | 25.4% | 4.5 | 14.1% | 8.6 |
| Depreciation | 5.8% | 1.0 | 3.3% | 2.0 |
| **Net Revenue from Operations** | **4.5%** | **0.8** | **34.0%** | **20.8** |
| Opportunity Cost—Capital | 4.5% | 0.8 | 2.4% | 1.5 |
| **Resource Rent (approximate)** | **0.1%** | **0.0** | **31.6%** | **19.4** |

Our calculated resource rent is hence an approximation. First, we acknowledge that there may be some intramarginal rents (IMRs) being generated due to fleet heterogeneity [33]. Any such IMR in the SAT SG fishery would imply a negative resource rent. In the Gulf, large resource rents (approx) have been generated since the introduction of the ITQs [9, 34]. In 2006, pre-ITQ, similar to the SAT SG fishery today, the substantial rent was non-existent. As the ITQ years coincided with a consolidation of vessels, it is unlikely that IMR increased during this time. Importantly, most of the different regulations in each fishery apply *uniformly* to all fishers in each fishery, i.e., they should not be a source of heterogeneity *within* the fishery.

We estimate the estimated resource rent in the GOM RF fishery between 2014–2016 in the broad range of $20 million per year or, in percent of revenue terms, over 30% of total revenue. In stark contrast, the SAT SG fishery seems to generate little or no resource rent. All the resource rents have dissipated due to the combination of an inability to limit costs or increase revenue when faced with the regulations. The difference between the two fisheries is 30% of revenue, i.e., a large fraction of total revenue. As we argued throughout the paper, the most likely culprit for this divergence in economic outcomes is the choice management regime. A sensitivity analysis on our assumptions would not change the central results due to their extreme divergence. Also, expanding the data to five years (2014–2018) makes no difference. In summary, in two fisheries with similar geography, biology, technology, and embedded in the same economic environment, the choice of management regime leads to very different economic outcomes. Specifically, the ITQ managed fishery generates substantial resource rent for society; on the order of a large fraction of revenue; while the traditionally managed fishery fails to capture most or all of the potential rents.

## Discussion

Rather than provide analysis of any specific management action, we have attempted to calculate the cumulative economic effects of the differing management regimes used in two otherwise very similar southeast U.S. reef fisheries. We considered attempting to estimate the loss of revenue in the SAT SG resulting from the volatility of landings (due to species-specific seasons) but decided it was beyond the scope of this research as it would need to be conducted on a species-by-species basis; with sometimes thin price and market data. Nonetheless, the gluts generated by the race-to-fish when seasons open and the frequent shutdowns undoubtedly contribute negatively to the market bargaining position of the SAT SG fishers. Research on the demand for the SAT SG species, especially on how the ex-vessel prices relate to locally landed species, could be used to better predict possible gains if the SAT SG fishery was to bring product to the market in a more rational manner.

Beyond the revenue loss, the SA fishery's resource rent dissipation is largely due to the larger-than-necessary fleet. A fleet that is larger than necessary incurs additional fixed cost and opportunity cost of capital. The burning of hundreds of thousands of gallons of additional fuel, necessitated by extra travel, dissipates millions of dollars in additional value while adding to US carbon emissions. Despite the warnings in much of the literature about the effects of ITQs on crew labor, the ratio of the number of crew days between the SAT SG and GOM RF fisheries is only 1.29 when standardized by poundage (Table 1) and much of that additional labor is a product of constantly traveling back and forth to port. The aggregate expenditure numbers also hide the fact that the GOM RF fishery uses less labor, but pays it more.

A known, but unquantified problem in the SAT SG fishery is regulatory discarding due to season closures and size limits. At its worst, fishers might be producing additional valuable product only to discard it. Any management changes that would allow fishers to keep even some of such catch—even if the quota is fixed—would have a disproportionally big economic

effect, as would any efforts by the fleet to work cooperatively to reduce bycatch [35]. The extent of these issues in the Gulf are likewise unknown, although ITQs are hardly free of discarding and high grading problems [36, 37].

Beyond its detrimental effects on economic profitability, there are additional problems with the SAFMC's consistent use of trip limits as a management tool. The effects of regulatory trip limits are also difficult to predict, as tightening or loosening trip limits causes changes in effort induced by the changes in trip efficiency [38]. Hence NOAA has to take a precautionary approach to fishing seasons, closing them prematurely sometimes lest quotas be exceeded and result in shorter seasons the following year. Following a presentation of our conclusions to the SAFMC, we were also told that owner-operators sometimes spend additional funds keeping crews on payroll during closed seasons, further subtracting from commercial profits. Such payments would not be accounted for in our trip-level analysis, and it is unclear if they would be reported on the annual, vessel-level survey. Finally, there are potential spillover effects from closed fisheries, putting potential strain on other fisheries in the region.

We have not discussed the recreational fishing fleet in this paper, but the GFMC has also experimented with a separate quota for the for-hire portion of that fishing sector in an effort to increase economic benefits [39, 40]. The SAFMC, in contrast, has not yet limited entry into for-hire fishing, let alone separated out management of it from the general recreational angler. We have also avoided discussing safety at sea issues, although there is evidence that the Gulf's management has resulted in improvements [41], or compliance rates [42].

We have also not discussed some of the potential negatives of the GMFC ITQ programs, leaving that for other authors to explore. This paper focuses on economic efficiency and rent returns as a measure of management success. The GFMC introduced ITQs into the Reef Fish fishery primarily to reduce overcapacity and eliminate derbies [9]. The SAFMC has other goals for the Snapper Grouper Fishery Management Plan, including allowing consistent access across all sectors and maximizing social and economic opportunities [43]. Researchers have raised issues with ITQs leading to a loss of local level community [44], increasing fishermen's dependence on particular fisheries [24], and causing employment loss [45]. Most troublesome, ITQs can cause wealth dissipation for non ITQ owners [46] and promote so-called "armchair fishing" [47], an accusation we have heard in about Gulf fisheries. These negative social effects may be present in the Gulf ITQ fisheries [48]. The older South Atlantic wreckfish ITQ may offer a less problematic alternative, shareholders can only lease quota to other shareholders and the fishery hence many of the same equity issues that may plague the Gulf [27].

## Conclusion

In summary, the SAFMC and GFMC take very different approaches to commercial fishing management. The GFMC has expanded ITQ management to most of the valuable commercial species in the GOM RF fishery, while the SAFMC continues to rely on traditional management. We find that the choice of fishery management regime shows up as a regime shift in the productivity and profitability of the fisheries. While many of our specific definitions or assumptions used in the derivation of these results could be adjusted or refined (according to each researcher's judgement and research question focus), such changes will not eliminate the huge advantage in terms of economic performance of ITQs over traditional management.

The decision to switch to ITQs or other catch shares is a politically charged one, and many of the criticisms—enrichment of the initial shareholders of quota at the expense of future ones (and of the public, who ultimately own the resource itself) and "armchair fishing" by shareholders who lease but do not fish—are serious and proven externalities of a different sort. Our analysis here shows that the decision to utilize a traditional management approach comes with

substantial economic consequences of its own. Trip limits, short seasons, and the resulting derbies—and the rational response of capital stuffing [49]—have resulted in a renewable resource being utilized in a way that does not capture its potential economic value.

## Supporting information

**S1 Fig. Annual, vessel-level economic survey instrument (mail survey).**
(TIF)

## Author Contributions

**Conceptualization:** Christopher Liese.

**Data curation:** Christopher Liese.

**Investigation:** Scott Crosson.

**Methodology:** Christopher Liese.

**Project administration:** Christopher Liese.

**Supervision:** Christopher Liese.

**Validation:** Christopher Liese.

**Visualization:** Scott Crosson.

**Writing – original draft:** Scott Crosson.

**Writing – review & editing:** Scott Crosson.

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
