## [Decision Letter · Decision Letter 0]

10 Apr 2023

PONE-D-22-30978Quantifying the economic effects of different fishery management regimes in two otherwise similar fisheriesPLOS ONE

Dear Dr. Crosson,

Thank you for submitting your manuscript to PLOS ONE. After careful consideration, we feel that it has merit but does not fully meet PLOS ONE’s publication criteria as it currently stands. Therefore, we invite you to submit a revised version of the manuscript that addresses the points raised during the review process.

Please consider all of the reviewers' comments. However, while one of the reviewers suggested that the manuscript would be improved if it were shorter (moving material to appendices or supplementary material), PLOS ONE does not have word limits for the body of a manuscript and length per se would not be a factor in whether the manuscript was accepted or not.

We look forward to receiving your revised manuscript.

Kind regards,

John A. B. Claydon, Ph.D.

Academic Editor

PLOS ONE

5. We note that [Figure 1] in your submission contain [map/satellite] images which may be copyrighted. All PLOS content is published under the Creative Commons Attribution License (CC BY 4.0), which means that the manuscript, images, and Supporting Information files will be freely available online, and any third party is permitted to access, download, copy, distribute, and use these materials in any way, even commercially, with proper attribution. For these reasons, we cannot publish previously copyrighted maps or satellite images created using proprietary data, such as Google software (Google Maps, Street View, and Earth). For more information, see our copyright guidelines: http://journals.plos.org/plosone/s/licenses-and-copyright.

Reviewers' comments:

Reviewer's Responses to Questions

**Comments to the Author**

1. Is the manuscript technically sound, and do the data support the conclusions?

Reviewer #1: Partly

Reviewer #2: Yes

2. Has the statistical analysis been performed appropriately and rigorously? 

Reviewer #1: Yes

Reviewer #2: Yes

3. Have the authors made all data underlying the findings in their manuscript fully available?

Reviewer #1: Yes

Reviewer #2: Yes

4. Is the manuscript presented in an intelligible fashion and written in standard English?

Reviewer #1: Yes

Reviewer #2: Yes

5. Review Comments to the Author

Reviewer #1: Comparing the Gulf and South Atlantic reef fish/ snapper-grouper fisheries in terms of how they are managed presents a unique opportunity to assess management regimes with regard to how they achieve economic efficiency and profitability. Based on previous research on ITQs, the authors assume that these are preferable systems because they return more economic value from a renewable natural resource. While this may be a sound assumption, to assess value based solely on profitability and efficiency (derived from calculating resource rents, or the profits that exceed business and opportunity costs) narrows the comparative value the authors seek to achieve. In addition to previous studies, the authors mention other assumptions about common property resources that have long been disputed in the literature, such Hardin’s 1950s assumption that common properties are always prone to overexploitation, which they characterize as “long-established.”

Privatizing fishing resources through ITQ systems may result in higher rents, but they may also reduce the extent to which those fisheries distribute those rents through the wider economy, which calls into question the value of achieving economic efficiency. Their own data on the sizes of the fisheries and the number of people employed suggest that the “inefficiency” of the South Atlantic fishery results in higher employment and, consequently, a wider distribution of the value of the resource, as wages/ shares move through local economies. Their wording that “the SAT SG fishery only uses 29% more labor than the GOM RF fishery” deemphasizes this, but 29% is close to one-third of the labor, which doesn’t seem insignificant. By contrast, the findings regarding the impacts of the different regimes on derby fishing are interesting and important, and these have been supported by the 5-year reviews of the ITQ programs.

Although I recognize these biases in the article, I do not advocate against publication, as long as the authors mention the social costs that many studies have found with ITQ management systems, particularly regarding equity and impacts on crew vs. owner-operators. As far as it goes, the economic analysis is sound, if somewhat misleading regarding the overall benefits of ITQ systems.

The authors would benefit from reading a recent (2021) Oceans Study Board report entitled, “The use of limited access privilege programs in mixed-use fisheries,” which reviews limited entry and ITQs from an interdisciplinary perspective and, among other things, finds that most ITQ systems emerge from limited entry systems and therefore are not as radically different from one another as the authors suggest. There have also been 5-year evaluations of both the red snapper and grouper-tilefish ITQ programs in the Gulf that the authors might want to read.

Reviewer #2: This paper evaluates the economic efficiency of two adjacent and ecologically similar fisheries in the southeast USA. The comparison produces useful and relevant results that should be of widespread interest as well as for the two management councils responsible for the fisheries. The data used and the methods appear to this reviewer to be sound.

There are, however, two important issues that should be addressed before the paper is suitable for publication. The first is the fundamental approach and style of the main argument of the paper. The comparison is between a fishery that is managed through ITQs (GOM) and one that is managed by a suite of regulations that the authors refer to as traditional regulations (SAT). The economic assessment demonstrates convincingly the substantially higher economic efficiency of the ITQ managed one and the reader is left with the impression that management is faced with a choice of one or other approach, and the solution is clearly that the SAT fishery should choose to follow the lead of the GOM.

The question of what is the best approach is not primarily a scientific issue and in evaluating the options and potential solution it is important to know and understand the reasons for the current arrangements in both fisheries. Why has SAT retained a ‘traditional’ approach, what are the objectives of the key stakeholders and to what extent does the current arrangement address or fail to address those objectives. The same questions need to be answered for the GOM fishery and, from the background to both, it should be possible to determine whether an ITQ system would be more or less successful in achieving the full set of SAT objectives than the current approach. For example, in lines 356-357 it is stated that “Regulations for most of the major commercial fishing species in the SAT SG fishery are intended to extend fishing opportunities”. If that is an important objective, is there a better approach for meeting it than either the current, economically inefficient system or a straightforward ITQ system? The authors cannot be expected to give definitive answers to this but they should give some attention to other important objectives in the SAT fishery and how the alternative management approaches would perform in achieving them (and, if relevant, whether there is any significant dissatisfaction in the GOM fishery as a result of the ITQ system).

In addition, and flowing from the assessment against the objectives, the choice of management approach is not simply a stark choice between one or the other. Unless it is clear that an ITQ system would satisfactorily meet the SAT’s objectives, there may be intermediate solutions to improving economic efficiency without going to full ITQs. Individual quotas don’t have to be transferable for example. In the opinion of this reviewer, the authors need to put their economic analyses into the context of the objectives in this way so that the current arrangements can be better understood and the results and authors’ conclusions to be more complete. Addressing this first issue need not be a major task and should not detract from the economic comparison, which is the primary purpose of the paper, but it is an important gap in the current version.

The second important issue is that the paper is too long and goes into too much detail in places in describing aspects of the methods and data collection, which distracts the reader with information that is not essential for understanding the problem and results. Much of that detail should go into either an appendix or as supplementary information.

Some other more specific comments are:

L35. The term ‘traditional fisheries management’ used here and elsewhere needs to be qualified because what is traditional in the US is not necessarily traditional elsewhere. They could simply state something like ‘In traditional fisheries management in the USA (subsequently referred to as traditional fisheries management)…’

L50. The authors need to make it clear here and elsewhere that where they refer to ‘rationalizes’, ‘efficiency’ etc, they are referring only to economic rationalizing etc. The consequences of ITQs may not lead to rationalizing on other objectives – a well known example is the decline in fishing and hence rural populations in more remote regions of Norway as a result of ITQs.

L91. Should be ‘Some - ’ or ‘Many economists…’ Not all economists assume minimizing costs is the sole objective.

L125 – 128. ‘Researchers have noted the effects, positive and sometimes negative of ITQs ….’ or something similar to make it clear that the papers referred to include both.

L215 and following. The authors should explain for both GOM and SAT how the total catch is regulated. Presumably in the GOM, a total allowable catch is set for each species or species assemblage, which is then divided into individual quotas according to the rights held by each holder. That, or if some other arrangement, should be explained. The likely approach in the SAT is less clear to readers unfamiliar with the fishery (including this one). Presumably in this case too, a total sustainable catch is determined each season, or periodically, and the lengths of the season, with the catch limits per trip, are used to restrict the total catch to that limit but, again, this or whatever the arrangement is, should be explained. Is the SAT fishery open access or are the numbers of permits limited (in which case what was the basis for setting the maximum number of permits allowed)?

L269. Explain or define ‘IFQ leasing’.

L344. Is there an explanation for why the SAT SG fishers receive a lower price per pound than the GOM fishers? Is it a result of the seasonal effect referred to elsewhere or are there other factors too?

L474 – 476. A reference or substantiation of some sort is needed for this statement.

L602-604. By omission these two sentences imply that discarding is not an issue in the GOM fishery. That would be very surprising as discarding and high-grading are problems in many other ITQ and IQ systems. The status in the GOM, in accordance with best available information, should also be stated here.

6. PLOS authors have the option to publish the peer review history of their article (what does this mean?). If published, this will include your full peer review and any attached files.

Reviewer #1: **Yes: **David Griffith

Reviewer #2: No

---

## [Author Response · Author response to Decision Letter 0]

12 May 2023

Dear PLOS ONE editors,

Attached is a copy of the newly revised paper, “Quantifying the economic effects of different fishery management regimes in two otherwise similar fisheries” (Manuscript Number: PONE-D-22-30978). We believe we have addressed the concerns raised by the PLOS ONE editorial staff and the manuscript reviewers. Our response to the major comments are below. 

PLOS ONE

“Please ensure that your manuscript meets PLOS ONE's style requirements, including those for file naming”

Done, and Figures have been moved into separate files.

“In your Data Availability statement, you have not specified where the minimal data set underlying the results described in your manuscript can be found. “

“We note that you have indicated that data from this study are available upon request. PLOS only allows data to be available upon request if there are legal or ethical restrictions on sharing data publicly.”

We added language into the document to this effect. Aggregated data and supporting documentation can be found at the NOAA repositories: https://repository.library.noaa.gov/. 

“Please include your full ethics statement in the ‘Methods’ section of your manuscript file. In your statement, please include the full name of the IRB or ethics committee who approved or waived your study, as well as whether or not you obtained informed written or verbal consent. If consent was waived for your study, please include this information in your statement as well.”

Our surveys are reviewed by the US Office of Management and Budget, which is our IRB-type process that follows US federal law. 

“We note that [Figure 1] in your submission contain [map/satellite] images which may be copyrighted. “

We have obtained and will upload the signed permission from the figure’s creator.

“Please review your reference list to ensure that it is complete and correct”

Done and revised according to PLOS ONE standards.

REVIEWER 1

“ the authors assume that these are preferable systems because they return more economic value from a renewable natural resource”

“Privatizing fishing resources through ITQ systems may result in higher rents, but they may also reduce the extent to which those fisheries distribute those rents through the wider economy”

“ as long as the authors mention the social costs that many studies have found with ITQ management systems, particularly regarding equity and impacts on crew vs. owner-operators.”

“Privatizing fishing resources through ITQ systems may result in higher rents, but they may also reduce the extent to which those fisheries distribute those rents through the wider economy”

Both reviewers noted that we stressed the importance of economic efficiency almost exclusively, and downplayed the importance of some of the negative impacts of ITQs. We acknowledge that and added language in the introduction and discussion sections (including a new paragraph) on some of the negative effects noted in the literature. We also make it clear that our focus is on the economic efficiency of management systems and should be received thusly.

“Note the 2021 Oceans Studies Board /NAS and Five Year Review of the Gulf tilefish/grouper ITQ program”

We added comments and cites accordingly.

“Their wording that “the SAT SG fishery only uses 29% more labor than the GOM RF fishery” deemphasizes this, but 29% is close to one-third of the labor”

A fair point and we replaced the phrase “relatively minor” with “lesser”,

REVIEWER 2

“question of what is the best approach is not primarily a scientific issue and in evaluating the options and potential solution it is important to know and understand the reasons for the current arrangements in both fisheries.”

“the choice of management approach is not simply a stark choice between one or the other. “

“L125 – 128. ‘Researchers have noted the effects, positive and sometimes negative of ITQs ….’ or something similar to make it clear that the papers referred to include both.”

“L50. The authors need to make it clear here and elsewhere that where they refer to ‘rationalizes’, ‘efficiency’ etc, they are referring only to economic rationalizing etc. The consequences of ITQs may not lead to rationalizing on other objectives – a well known example is the decline in fishing and hence rural populations in more remote regions of Norway as a result of ITQs.”

We added language acknowledging this, including a paragraph of some of the detrimental social effects of ITQs, as well as citations from both Councils regarding the management goals of the ITQs or Fishery Management Plans. See our comments to Reviewer 1. We also point out that the sole ITQ in the South Atlantic region (wreckfish) may provide a middle ground between the current management and the relatively unrestricted ITQ in the Gulf.

“ lines 356-357 it is stated that “Regulations for most of the major commercial fishing species in the SAT SG fishery are intended to extend fishing opportunities””

We provided more details and a citation on this in the conclusion after help from Council staff.

“L35. The term ‘traditional fisheries management’ used here and elsewhere needs to be qualified because what is traditional in the US is not necessarily traditional elsewhere. They could simply state something like ‘In traditional fisheries management in the USA (subsequently referred to as traditional fisheries management)”

L91. Should be ‘Some - ’ or ‘Many economists…’ Not all economists assume minimizing costs is the sole objective.

L215 and following. The authors should explain for both GOM and SAT how the total catch is regulated. Presumably in the GOM, a total allowable catch is set for each species or species assemblage, which is then divided into individual quotas according to the rights held by each holder. That, or if some other arrangement, should be explained. The likely approach in the SAT is less clear to readers unfamiliar with the fishery (including this one).

L269. Explain or define ‘IFQ leasing’.”

These minor edits were completed, thank you.

“L474 – 476. A reference or substantiation of some sort is needed for this statement.”

We edited the statement to better reflect our focus on the spread between returns in the two regions. 

“L602-604. By omission these two sentences imply that discarding is not an issue in the GOM fishery. That would be very surprising as discarding and high-grading are problems in many other ITQ and IQ systems. The status in the GOM, in accordance with best available information, should also be stated here.”

Good point and we acknowledge that in the revised paper along with citations. 

“L344. Is there an explanation for why the SAT SG fishers receive a lower price per pound than the GOM fishers? Is it a result of the seasonal effect referred to elsewhere or are there other factors too?”

We are uncertain to what extent seasons or other factors are causing this price difference, although we are aware of ongoing research investigating it right now. 

“paper is too long and goes into too much detail in places”

It is definitely a long paper, but we wanted a thorough investigation of these issues, and chose PLOS ONE for its tolerance for somewhat long pieces with multiple graphics.

---

## [Editor Report · Decision Letter 1]

2 Jun 2023

Quantifying the economic effects of different fishery management regimes in two otherwise similar fisheries

PONE-D-22-30978R1

Dear Dr. Crosson,

We’re pleased to inform you that your manuscript has been judged scientifically suitable for publication and will be formally accepted for publication once it meets all outstanding technical requirements.

Kind regards,

John A. B. Claydon, Ph.D.

Academic Editor

PLOS ONE
---

## [Editor Report · Acceptance letter]

9 Jun 2023

PONE-D-22-30978R1 

Quantifying the economic effects of different fishery management regimes in two otherwise similar fisheries 

Dear Dr. Crosson:

I'm pleased to inform you that your manuscript has been deemed suitable for publication in PLOS ONE. Congratulations! Your manuscript is now with our production department. 

Kind regards, 

on behalf of

Dr. John A. B. Claydon 

Academic Editor

PLOS ONE